# The Application of Optical Genome Mapping (OGM) in Severe Short Stature Caused by Duplication of 15q14q21.3

**DOI:** 10.3390/genes14051016

**Published:** 2023-04-29

**Authors:** Xiaoan Ke, Hongbo Yang, Hui Pan, Yulin Jiang, Mengmeng Li, Hanzhe Zhang, Na Hao, Huijuan Zhu

**Affiliations:** 1State Key Laboratory of Complex Severe and Rare Diseases, Chinese Research Center for Behavior Medicine in Growth and Development, Department of Endocrinology, Peking Union Medical College Hospital, Chinese Academy of Medical Science and Peking Union Medical College, Beijing 100730, China; 2Department of Obstetrics and Gynecology, National Clinical Research Center for Obstetric and Gynecologic Diseases, Peking Union Medical College Hospital, Chinese Academy of Medical Sciences and Peking Union Medical College, Beijing 100730, China

**Keywords:** optical genome mapping (OGM), short stature, genomic structural variants (SVs), duplication

## Abstract

(1) Background: Optical genome mapping (OGM) is a novel approach to identifying genomic structural variations with high accuracy and resolution. We report a proband with severe short stature caused by 46, XY, der (16) ins (16;15) (q23; q21.3q14) that was detected by OGM combined with other tests and review the clinical features of patients with duplication within 15q14q21.3; (2) Methods: OGM, whole exon sequencing (WES), copy number variation sequencing (CNV-seq), and karyotyping were used; (3) Results: The proband was a 10.7-year-old boy with a complaint of severe short stature (−3.41SDS) and abnormal gait. He had growth hormone deficiency, lumbar lordosis, and epiphyseal dysplasia of both femurs. WES and CNV-seq showed a 17.27 Mb duplication of chromosome 15, and there was an insertion in chromosome 16 found by karyotyping. Furthermore, OGM revealed that duplication of 15q14q21.3 was inversely inserted into 16q23.1, resulting in two fusion genes. A total of fourteen patients carried the duplication of 15q14q21.3, with thirteen previously reported and one from our center, 42.9% of which were de novo. In addition, neurologic symptoms (71.4%,10/14) were the most common phenotypes; (4) Conclusions: OGM combined with other genetic methods can reveal the genetic etiology of patients with the clinical syndrome, presenting great potential for use in properly diagnosing in the genetic cause of the clinical syndrome.

## 1. Introduction

With recent developments in testing and the standardization of evaluation procedures, more and more genetic causes of short stature are being identified [1,2]. One study revealed that approximately 50% of severe short-stature cases in children might have a genetic etiology [3]. However, the rate of identification of the causative genetic factors in short stature below −3SD is still less than 30% [4]. Genomic structural variants (SVs) are common events that are associated with many specific phenotypes and diseases [5], which is noteworthy because aberrations in chromosomes and chromosomal instability have been reported in patients with clinical syndrome [6,7], such as Silver–Russell Syndrome [8], Turner syndrome [9], and Prader–Willi Syndrome [10].

Traditional detection techniques for genomic SVs include karyotyping, fluorescence in situ hybridization (FISH), and copy number variation (CNV) microarrays. Each of these methods has its own limitations [11], however. Optical genome mapping (OGM) is one approach to analyzing large genomes and structural features at a high resolution. Recently OGM has become a highly promising method for detecting large-scale SVs due to its high accuracy and concordance with more traditional methods [12]. In this study, we report for the first time a proband with severe short stature caused by 46, XY, der (16) ins (16;15) (q23; q14q21.3) detected by OGM combined with other genetic testing and review the clinical features of patients with duplication within 15q14q21.3. Thus, this paper reveals the potential role of OGM in identifying the genetic cause of patients with clinical syndrome.

## 2. Materials and Methods

### 2.1. Participants

This study only includes the proband from our medical center and his family members. The proband patient underwent evaluation for short stature, including testing of liver and kidney function, thyroid function, gonadal function, serum adrenocorticotropin levels and serum cortisol levels, and serum insulin-like growth factor-1 (IGF-1) levels. A growth hormone-levodopa stimulation test and an insulin tolerance test (ITT) were also performed (5 blood collections in 0 min, 30 min, 60 min, 90 min, and 120 min). X-ray was used to assess the spine, pelvis, and bone age, and clinical information, including birth history, growth and development history, present history, and family history, was collected. The study was approved by the institutional review board (IRB) of Peking Union Medical College Hospital.

### 2.2. Whole-Exome Sequencing (MGISEQ-2000)

To carry out whole-exome sequencing (WES), two milliliters of peripheral blood were drawn from the proband and his family, and genomic DNA (gDNA) was extracted according to the manufacturer’s instructions (D3537-02, MAGEN, Guangzhou, China). The gDNA was broken into fragments using BGI’s enzyme kit (Segmentase; BGI), and a single individual DNA library was constructed after LM-PCR and purification. Next, sequencing was performed using the PE100 + 100 set in MGISEQ-2000. To detect potential variants, we performed bioinformatics processing and data analysis after receiving the primary sequencing data and used previously published filtering criteria to generate “clean reads” for further analysis [13]. The “clean reads” (90 bp in length) derived from targeted sequencing and filtering were then aligned to the human genome reference (hg19) using the BWA (Burrows–Wheeler Aligner) Multi-Vision software package (BWA-0.7.17, r1188) [14]. After alignment, the output files were then used to perform sequencing coverage and depth analysis of the target region, single-nucleotide variants (SNVs), and INDEL calling, where GATK software (4.1.9.0) [15] was used to detect SNVs and indels. All the SNVs and indels were then filtered and estimated via multiple databases, including NCBI dbSNP, HapMap, the 1000 human genome dataset, and the database of 100 healthy Chinese adults.

### 2.3. Copy Number Variation Sequencing (CNV-Seq)

For CNV-seq, DNA was extracted from peripheral blood using a MagPure Buffy Coat DNA Midi KF kit (Magen, Guangzhou, China), followed by DNA fragmentation, library construction using PCR technology, and index addition. The quality of the libraries was tested using a Qubit dsDNA HS Assay kit (Invitrogen, Waltham, MA, USA) before and after pooling the libraries, and the hybrid library was then subjected to 35 bp single-terminal sequencing with a sequencing depth of 0.41× using a DNBSEQ-G400 High-throughput Sequencing Set and the MGISEQ-2000 sequencer (MGI, Shenzhen, China). The results were determined by referring to the hg19 version of the human genome and the most recent data available on the Database of Genomic Variants, DECIPHER, Online Mendelian Inheritance in Man, University of California Santa Cruz, PubMed, ClinGen, DGV, and other public databases. Finally, the clinical significance of CNVs was divided into five classes according to the American College of Medical Genetics (ACMG) sequence variant classification guidelines for copy number variants [16]: pathogenic, likely pathogenic, benign, likely benign, and variant of uncertain significance (VUS).

### 2.4. Optical Genome Mapping (OGM)

Bionano optical genome mapping was performed as described previously in the literature [17,18]. First, ultra-high-molecular-weight (UHMW) DNA was extracted from whole blood following the Bionano Prep SP Fresh Human Blood DNA Isolation Protocol v2 (Bionano Genomics, San Diego, CA, USA). Briefly, the CTTAAG motif on DNA was fluorescently labeled with green fluorescence, and stained DNA backbone was stained using a Direct Label and Stain (DLS) kit (Bionano Genomics, USA) according to the manufacturer’s instructions. Labeled DNA was quantified to 4–12 ng/uL using a Qubit dsDNA HS Assay kit and Qubit Fluorometer (ThermoFisher Scientific, Waltham, MA, USA). Then, 20 uL labeled DNA was loaded on a Saphyr chip (Bionano Genomics) for linearization and imaged on a Saphyr instrument (Bionano Genomics). A Data QCBioinformatics analysis de novo assembly pipeline was performed using Bionano Access v1.7, and Bionano Solve v3.7, including de novo assembly, alignment with the genome reference (hg19GRCh 37), and identification of SVs.

The list of genes and their function in the duplication of chr15 were obtained from genome NCBI. GO enrichment and KEGG pathway analysis of the genes in the duplication was realized through Metascape (http://metascape.org accessed on 13 April 2023).

### 2.5. Long-Range PCR and Sanger Sequencing

Long-range PCR was used to identify the exact breakpoint sites of chromosome SVs, and the PCR reaction was performed in a 50 µL volume that included 10 µL 5 × PrimeSTAR GXL Buffer, 4 µL dNTP Mixture (2.5 mM each), 1 µL TaKaRa Prime STAR GXL DNA Polymerase, 1 µL each of forward and reversed primers (10 µM), 2 µL DNA template and 31 µL of PCR-grade H_2_O under the following parameters: 94 °C for 1 min, followed by 35 cycles of amplification (denaturation 98 °C, 10 s; annealing 50 °C, 15 s, extension 68 °C, 10 min). PCR products were then detected on 0.8% 1 × TAE agarose gels and sequenced using Illumina platforms, and target region flanking fusion sites were further validated by Sanger sequencing with an amplified fragment.

### 2.6. Literature Review and Database Search

We searched for previously published cases of duplication within 15q14q21.3 duplication on PubMed, Embase, and Medline databases (up to 4 October 2022, in English) using “15q duplication”, “15q15 duplication”, “15q21 duplication”, and “15q duplication syndrome” as search terms. We also searched for all reported cases of duplication within 15q14q21.3 on the University of California Santa Cruz (UCSC, http://genome.ucsc.edu/cgi-bin/hgGateway accessed on 25 September 2022) database. All pathogenic or likely pathogenic gain coverages in DECIPHER CNVs [19] and ClinGen CNVs [20] from UCSC were included (GRCh37/hg19).

## 3. Results

### 3.1. Case Presentation

A 10.7-year-old boy with a complaint of short stature and abnormal gait was first referred to our clinic. He was the third child in his family and had a full-term gestation period and a normal birth history. His birth weight was 3.25 kg, and his birth length was unknown. The patient’s gait was found to be abnormal when he was three years old, and hip valgus was discovered at age four. The patient was shorter than his peers at an early age, and the annual growth rate was 5 to 6 cm every year. Height increased by 3 cm in the most recent year, and no secondary sexual characteristics were observed. His intelligence was well-developed.

Initial physical examination revealed that the patient’s height was 120.7 cm (height SDS: −3.41), and his weight was 29.5 kg (weight SDS: −0.81). His lower extremities are shorter than average, indicating a disproportionate short stature. The patient’s growth chart is shown in Figure 1a. The patient also had a short and flat nose with no high-arched palate or other facial abnormalities, and there was a brown spot on his left abdominal wall. In terms of family history, the heights of his family members are shown in Figure 1b. The patient’s elder brother and sister were both adults with normal growth development.

The etiology of short stature was evaluated, including hormonal tests. The peak value of growth hormone (GH) was 4.34 ng/mL and 5.01 ng/mL by growth hormone-levodopa stimulation test and insulin tolerance test (ITT), revealing growth hormone deficiency (GHD). Serum IGF-1 levels were 261 ng/mL (normal range 88–452 ng/mL), and serum thyroid stimulating hormone (TSH), free triiodothyronine (FT3), and free thyroxine (FT4) levels were 1.914 uIU/mL (normal range: 0.380 to 4.340 uIU/mL), 3.47 pg/mL (normal range:1.80 to 4.10 pg/mL) and 1.14 ng/dl (normal range: 0.81–1.89 ng/dl), respectively. For sex hormones, luteinizing hormone (LH) was 0.21 IU/L, follicle-stimulating hormone (FSH) was 2.95 IU/L, and testosterone (T) was below 0.1 ng/mL, giving no sign of puberty development.

Additionally, the patient’s bone age was consistent with his chronological age. Echocardiography revealed mild to moderate tricuspid regurgitation, and the X-ray showed lumbar lordosis and epiphyseal dysplasia of both femurs (Figure 1c,d). The patient was 12.7 years old at this last follow-up, and his height had increased to 8.4 cm in two years for an annual growth rate of 4.2 cm per year. However, it was still the case that no secondary sexual characteristics were observed. Thus, the proband presented with severe short stature with abnormal skeletal dysplasia, mild to moderate tricuspid regurgitation, and GHD.

### 3.2. A 17.27 Mb Duplication of the Long Arm of Chromosome 15 and an Insertion at the Long Arm of Chromosome 16

After clinical evaluation for short stature, WES was conducted to explore possible genetic causes. Although no candidate pathogenic single nucleotide variants were identified, a 17.27 Mb duplication of chromosome 15 (15q14-q21.3) was discovered and confirmed by CNV-seq (Figure 2a). There were 309 genes in this duplication, and 167 of them were protein-coding genes, the role of which were described in the Appendix A. GO analyses revealed that these genes were mainly enriched in the following processes: microtubule cytoskeleton organization, phosphatidylglycerol acyl-chain remodeling, chromosome segregation, vesicle organization, etc. (Appendix A). The KEGG pathways enrichment showed that these genes were mainly enriched in three pathways: ovarian steroidogenesis, thyroid hormone synthesis, and arginine and proline metabolism (Appendix A). In addition, during the routine workup, karyotyping revealed an insertion in chromosome 16 with a diagnosis of 46, XY, der (16)t (16;?) (q23;?), as shown in Figure 2b. However, the parents were free of the above structural variation. Thus, a de novo duplication of 15q14q21.3 and an insertion at the long arm of chromosome 16 were found in the proband by three separate genetic tests.

### 3.3. A Duplication of 15q14q21.3 Inversely Inserted to 16q23.1 by OGM

In an attempt to confirm and characterize the SV found by WES, CNV-seq, and karyotyping in the proband, we carried out OGM. As shown in Figure 3, there was an inter-translocation (inversion) between chromosomes 15 and 16, and a duplicative copy number of chromosome 15, 40.04–57.29 Mb, was also found. In addition, OGM revealed that there were two breakpoints in chromosome 15 overlapping with the *TCF12* gene and *FSIP1* gene and one breakpoint in chromosome 16 overlapping with the *CFDP1* gene (Figure 4). The 75.399 Mb (Chr 16: 75399363 bp) of chromosome 16 was connected to the 57.297 Mb (Chr 15: 57297589 bp) of Chromosome15 to form a *TCF12-CFDP1* fusion gene (breakpoint one), which was determined to retain the *CFDP1* promoter (Figure 4a,b). In addition, the 40.044 Mb of Chromosome 15 (Chr 15: 40044538 bp) was connected to the 75.405 Mb (Chr 16: 75405351 bp) of Chromosome 16 to form an *FSIP1*-*CFDP1* fusion gene (breakpoint two, Figure 4c). These two breakpoint sites caused by the rearrangement were confirmed by LR-PCR, NGS, and Sanger sequencing (Appendix A). There were other variants in the circos plot; 1% in the control database, meanwhile overlapping with OMIM genes (Bionano Access 1.7 software built-in), was used to filter the variants. The remaining included five insertions, four deletions, and four duplications; the detailed information on the variants and involved genes are listed in Appendix A. Of note, no variant on that list was related to the clinical phenotypes of our proband. Thus, structure variation, 46, XY, der (16) ins (16;15) (q23; q21.3q14) dn, was carried by our patient and may be responsible for his clinical syndrome.

### 3.4. Clinical Features of Patients with Duplication within the 15q14q21.3 Region

After searching the databases, a total of fourteen patients had duplication within the 15q14q21.3 region (Table 1), with thirteen previously reported and one from our center. About half of them (6/14, 42.9%) were de novo, and one patient’s condition had been maternally inherited. Our patient was the only one with an invertedly inter-translocation, however. The lengths of these duplications were from 65.6 Kb to 17.2 Mb, and the structural variations were diagnosed primarily by clinical cytogenomic testing, including single-nucleotide polymorphism microarray, chromosomal microarray analysis, and high-resolution G-banded cytogenetic analysis. We further summarized the phenotypes of patients with duplication within 15q14q21.3 in Table 2. The most common phenotype of these patients was neurologic symptoms (10/14, 71.4%) that mainly manifested as mental and developmental retardation. This was followed by abnormal growth (7/14, 50.0%), facial (6/14, 42.9%) and skeletal abnormalities (5/14, 35.7%).

## 4. Discussion

In this study, we reported one severely short-stature patient with the clinical syndrome who we found to have 15q14q21.3 duplication and insertion in chromosome 16 by WES, karyotyping, and CNV-seq. Further, OGM revealed the specific information of the structural variation as 46, XY, der (16) ins (16;15) (q23; q21.3q14) dn. In addition, other recorded patients with duplication within 15q14q21.3 were mainly de novo, and the most common phenotypes were neurologic symptoms (10/14, 71.4%), abnormal growth (7/14, 50.0%), and facial (6/14, 42.9%) abnormalities.

Many children classified under idiopathic short stature (ISS) have underlying genetic causes [21], and 50% of severe short stature falls under this category [3]. In these circumstances, clinicians may employ genetic testing, such as karyotyping for Turner syndrome, chromosomal microarray for chromosomal structural abnormalities, targeted gene panels for suspected genes, and WES for genetic screening [21]. However, the rate of identifying the causative genetic anomalies using these genetic approaches is still low (~30%), especially with WES [22,23]. One previous study demonstrated that 5–10% of patients with short stature have copy number variants that are highly likely to be pathogenic, giving a potential total diagnostic yield of only 10–20% [2].

So far, the diagnostic cytogenetic analysis includes mostly karyotyping, fluorescence in situ hybridization (FISH), and copy number variant (CNV) microarrays and each of these tests suffers from its own limitations [11]. Molecular genetic testing, such as WES, is also capable of detecting SVs; however, long regions of repetitive sequences in the human genome tend to be difficult or impossible to analyze using only short molecules of DNA [24,25].

In recent years, optical genome mapping (OGM) has been employed for clinical analysis to detect chromosomal numerical aberrations and SVs [12]. In 2021, a study by Mantere [26] et al. showed that high-resolution OGM reached 100% concordance compared to standard assays for 99 chromosomal aberrations with non-centromeric breakpoints. Moreover, another study by Neveling [11] et al. reported that OGM offered a more comprehensive assessment than any previous single test, and the authors also reported the most accurate underlying genomic architecture to date for 52 hematological malignancy genomes. Thus, the excellent concordance, the high-resolution, and accurate architecture of OGM with standard diagnostic assays showcase its potential to replace classical cytogenetic tests.

At present, OGM has been applied in the detection of many genetic diseases, such as Marfan syndrome [18], Duchenne muscular dystrophy (DMD) [27], and Digeorge deletion syndrome [28]. In our study, a duplication of 15q14q21.3 and an insertion in chromosome 16 were identified by CNV-seq and karyotyping, respectively. Additionally, OGM can reveal more accurate information about the inter-translocation between chromosomes 15 and 16, and in our case, two fusion gene results from the breakpoints were found by OGM. Thus, OGM offers the detection of structural variations with high resolution, which is a promising application in identifying the genetic structural variations of patients with this clinical syndrome.

There have been many previous studies on duplication of the long arm of chromosome 15 that have mainly included the proximal and distal segments. The features of chromosome 15q11-q13 duplication syndrome (OMIM 608636), which has been implicated in Angelman syndrome and Prader–Willi syndrome, include autism, mental retardation, ataxia, seizures, developmental delays, and behavioral problem. Many investigators have also reported several patients carrying duplication of the distal long arm in chromosome 15 [29,30,31], which manifests primarily as frontal bossing, short palpebral fissures, long philtrum, low-set ears, high-arched palate, retrognathia, arachnodactyly, microcephaly, joint contractures, and development delay. However, patients with duplication in the medial long arm of chr 15 are rare [32,33,34].

**Table 1 genes-14-01016-t001:** Review of partial duplication in 15q14q21.3.

No.	Gender	Age	Bands (Location)	Length	Origin	Methods	Phenotypes
Patient 1 [32]	boy	12 y	15q14.1q21.1 (NA)	15 Mb	De novo	High-resolution G-banded cytogenetic analysis	Distinctive face including a narrow forehead, high nasal bridge, narrow palate, dental malocclusion, small ears, and micrognathia; seizure; short stature; cryptorchidism, scrotal hypoplasia, and micro-penis; mental and developmental retardation; others: lumbar kyphosis, bell-shaped chest, mild scoliosis
Patient 2 [33]	boy	14 y	15q15.3q21.2 (chr15: 44,143,547–50,572,601)	6.4 Mb	De novo	Chromosomal microarray analysis	Distinctive facial features including macrocephaly, broad forehead, deep-set and widely spaced eyes, broad nose bridge, shallow philtrum, and thick lips; severe short stature; delayed bone age; endocrine: hypogonadism, micro-penis, small testes; global developmental delay and intellectual disability
Patient 3 [34]	boy	8 y	15q21.2 (chr15: 50,382,769–51,568,204)	1.1 Mb	De novo	Single-nucleotide polymorphism microarray	Gynecomastia; tall stature and bone age advancement
our patient	boy	10.7 y	15q14q21.3 (chr15: 40,044,538–57,297,589)	17.2 Mb	De novo	Karyotype, WES, CNV-seq, and OGM	Severe short stature; distinctive face: a short and flat nose; mild to moderate tricuspid regurgitation; lumbar lordosis and epiphyseal dysplasia of bilateral femur; growth hormone deficiency
nssv 3395697	-	-	15q21.1 (chr15: 48,728,235–48,793,803)	65.6 Kb	NA	Clinical Cytogenomic Testing (Postnatal)	Developmental delay AND/OR other significant developmental or morphological phenotypes
nssv 575513	-	-	15q21.1 (chr15: 48,728,235–48,793,803)	65.6 Kb	NA	Clinical Cytogenomic Testing (Postnatal)	Global developmental delay
nssv 578686	-	-	15q14q15.1 (chr15: 36,824,194–41,079,736)	4.256 Mb	NA	Clinical Cytogenomic Testing (Postnatal)	Developmental delay AND/OR other significant developmental or morphological phenotypes
nssv 578687	-	-	15q21.3 (chr15: 55,571,230–55,914,205)	0.343 Mb	NA	NA	Abnormal facial shape, Abnormal heart morphology, Developmental delay AND/OR other significant developmental or morphological phenotypes, Gastroschisis, Global developmental delay
Decipher 342055	-	-	15q21.3 (chr15: 54,534,868–55,384,248)	0.849 Mb	De novo	Microarray	Global developmental delay
Decipher 331008	-	-	15q15.1q15.2 (chr15: 42,621,710–43,056,143)	0.434 Mb	Maternally inherited	Microarray	Abnormal facial shape, tetralogy of Fallot
Decipher 308278	-	-	15q21.1 (chr15: 47,384,497–48,918,698)	1.534 Mb	NA	Microarray	Disproportionate tall stature
Decipher 401711	-	-	15q15.2q21.2 (chr15: 43,325,133–51,471,260)	8.146 Mb	De novo	Microarray	Delayed skeletal maturation, hypertelorism, intellectual disability, large earlobe, Macrotia, Pes planus, Proportionate short stature, Proptosis, Thick eyebrow, up-slanted palpebral fissure, Wide mouth
Decipher 303564	-	-	15q15.3q21.1 (chr15: 44,792,878–45,568,844)	0.776 Mb	NA	Microarray	Cognitive impairment
Decipher 385178	-	-	15q15.3q21.1 (chr15: 43,851,578–48,145,280)	0.385 Mb	NA	Microarray	Delayed ability to walk, short stature

Note: NA, unavailable.

In our study, patients with duplication in 15q14q21.3 differed significantly from one another, but they shared the common feature of mental and developmental retardation, which was consistent with duplication in the proximal and distal segments of 15q. Moreover, about half of them were de novo (6/14, 42.9%), although the inheritance information for seven patients was unavailable. In addition, this duplication can also manifest as distinctive facial features, skeletal abnormities, genitourinary dysplasia (cryptorchidism, micro-penis, and small scrotum), tetralogy of Fallot, and gastroschisis.

Nevertheless, the proband in our study only presented with severe short stature with skeletal abnormalities, mild to moderate tricuspid regurgitation, and GHD without mental or developmental retardation, which was inconsistent with previous studies [32,33]. We noticed that the breakpoints in our proband located in chromosome 16 resulted in two fusion genes, *TCF12*-*CFDP1* and *FSIP1*-*CFDP1*, and there were no associated phenotypes of *CFDP1* and *FSIP1* in OMIM. We additionally found that hypogonadotropic hypogonadism-26 with or without anosmia (HH26) was caused by heterozygous or homozygous mutation in *TCF12* (600480) on chromosome 15q21. However, the breakpoints in previously-reported cases were in chromosome 15 [32,33]. Thus, a possible explanation for our findings is that the discrepant phenotype of our patient and those in the literature may result from different breakpoints.

**Table 2 genes-14-01016-t002:** Clinical features of partial duplication in 15q14q21.3.

Phenotypes	N/Total (%)
Neurologic	10/14 (71.4)
Growth	7/14 (50.0)
Short stature	5/7 (71.4)
Tall stature	2/7 (28.6)
Facial abnormalities	6/14 (42.9)
Skeletal abnormalities	5/14 (35.7)
Genitourinary	3/14 (21.4)
Cardiovascular	3/14 (21.4)
Endocrine disorders	2/14 (14.3)
Abdomen	1/14 (7.1)

According to a joint consensus recommendation of the American College of Medical Genetics and Genomics (ACMG) and the Clinical Genome Resource (ClinGen) [16], the chromosomal structural variation involved 17.2 Mb in our proband, much larger than 5 Mb, was considered to be a pathogenic variant. To further exploration, we identified candidate genes responsible for clinical phenotypes. The duplication of 15q14q23.1 contained 309 genes (Appendix A), and 167 of them were protein-coding genes. GO enrichment showed these genes were mainly enriched in microtubule cytoskeleton organization, phosphatidylglycerol acyl-chain remodeling, and chromosome segregation. Additionally, there were 56 genes with phenotypes in OMIM.

It has been proposed that the three-dimensional structure of fibrillin molecules (product of FBN1) within a microfibril may create a microenvironment consisting of domains that cause acromelic dysplasia [35]. Fibrillin-1 (FBN1) is a major glycoprotein of the extracellular matrix, which contains repeating calcium-binding epidermal growth factor (cbEGF)-like domains interspersed with eight-cysteine (TB) domains. Disease-causing mutations of *FBN1* disrupting heparin binding by TB5 can result in Weill–Marchesani syndrome (WMS) or Acromicric (AD) and Geleophysic Dysplasias (GD) [36]. Distinguishing GD features are progressive thickening of the cardiac valves, toe walking, and a “happy” face characterized by full cheeks, a short nose, and a thin upper lip [37]. Moreover, duplication of the *FBN1* gene was a possible cause of skeletal abnormalities in the patient with 15q15.3q21.2 duplication has been reported by Yuan et al. [33], but the specific mechanism remained unknown. Indeed, the phenotypes of *FBN1*, located in 15q21.1, are related to severe short stature, skeletal dysplasia, toe walking, normal intelligence, and distinctive facial features, consistent with those of our proband. In addition, we searched the function and phenotypes of all the genes in 15q14q21.3, and only the *FBN1* gene could explain the complex clinical phenotypes of our proband. Therefore, we speculate that the structural variation may be responsible for the clinical phenotypes of our proband, and *FBN1* was the candidate gene. A previous study has shown a poor efficacy of recombinant human growth hormone (rhGH) treatment in patients with *FBN1* heterozygous mutation [38]. Considering the skeletal dysplasia and poor efficacy of rhGH treatment in patients with *FBN1* mutation, we did not treat our patient with rhGH.

In addition, the genes involved in this duplication are associated with other functions. *DUOX2* and *DUOXA2* are essential for the maturation and function of the DUOX enzyme complexes involved in thyroid hormone synthesis. Mutations in these two genes may result in thyroid dysfunction [39]. *CYP19A1* encodes a member of the cytochrome P450 superfamily of enzymes. This protein localizes to the endoplasmic reticulum and catalyzes the last steps of estrogen biosynthesis. Mutations in this gene can result in either increased or decreased aromatase activity [40]. The mutations in the *USP8* gene are closely related to the development of pituitary adrenotropic hormone-secreting pituitary adenomas [41]. Close monitoring is required in future clinical follow-ups.

Although this study is the first to report that OGM shows favorable accuracy and consistency with other genetic testing in the genetic diagnosis of patients with severe short stature, the study still has several limitations. First, OGM is rarely used in clinical practice, and more cases are needed to validate its feasibility and applicate in clinical practice. Second, the sample of patients with duplication within 15q14q21.3 was small, and more patients are needed to further summarize the clinical features. Third, the patient’s clinical data were partly missing, so a comprehensive clinical evaluation could not be performed.

## 5. Conclusions

OGM, combined with other molecular genetic technologies, can help to reveal the genetic etiology of patients with clinical syndrome. In addition, the duplication in 15q14q21.3 presents primarily with mental and developmental retardation, distinctive facial features, and skeletal abnormities, so genetic testing for structural variations in chromosomes is required for patients with these phenotypes.

## Figures and Tables

**Figure 1 genes-14-01016-f001:**
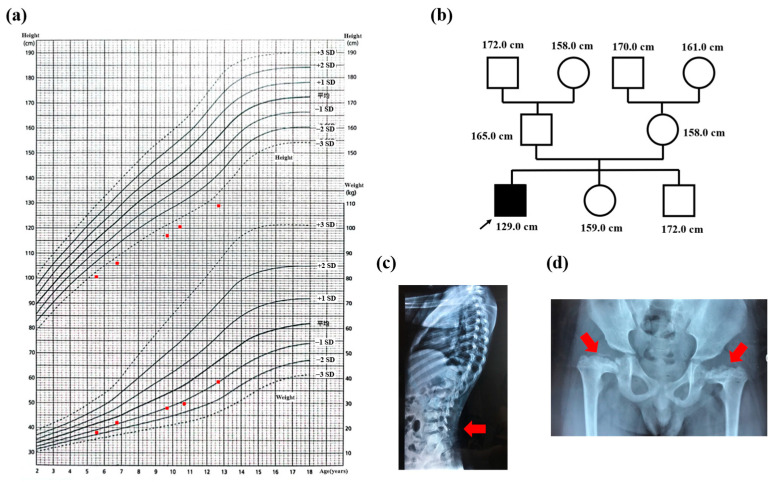
The clinical information of the proband. (**a**) the growth chart of the proband, the red dots indicated height and weight at different time points; (**b**) the height in his pedigree; (**c**) the lumbar lordosis in X-ray with red arrow; (**d**) the epiphyseal dysplasia of bilateral femur in X-ray with red arrow.

**Figure 2 genes-14-01016-f002:**
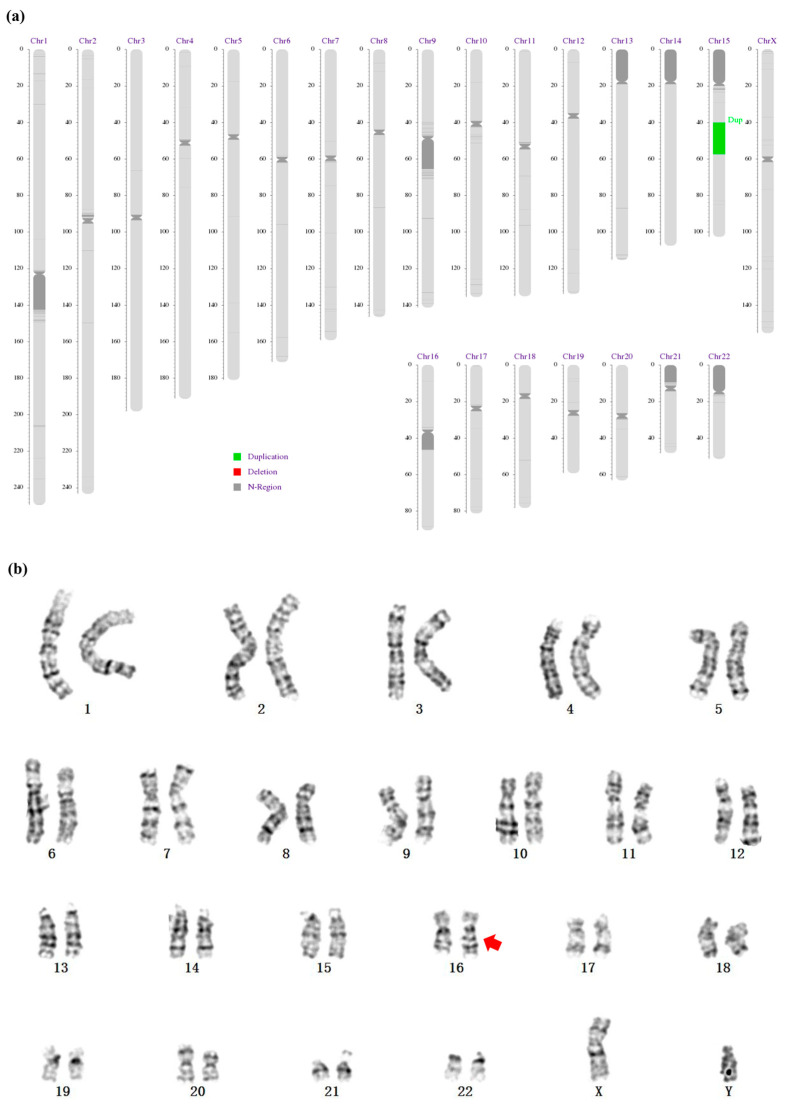
The duplication of chr15 found by CNV-seq ((**a**), green highlighted), and the insertion at chr16 found by karyotype ((**b**), red arrow).

**Figure 3 genes-14-01016-f003:**
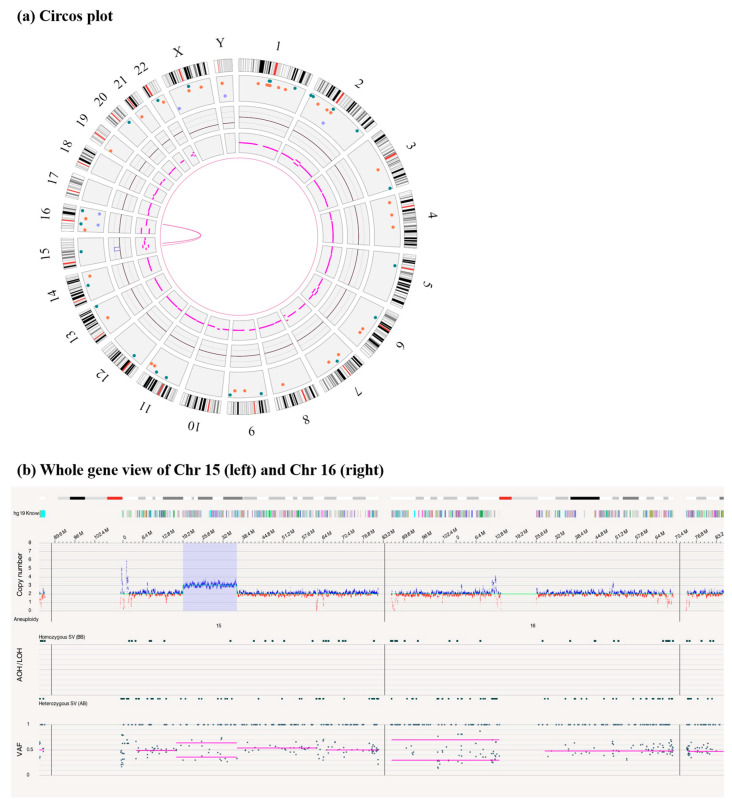
The information of structural variations and copy number variant by OGM. (**a**) Genome-wide circos plot showing t(15;16)(q14;q23.1). (**b**)The CNV profile showing the green line as the baseline of copy number equal to 2, the blue block as the copy number gain of 15q14q21.3 with size of 17.2 Mb (Chr 15: 40044538-57297589 bp), copy number = 3.

**Figure 4 genes-14-01016-f004:**
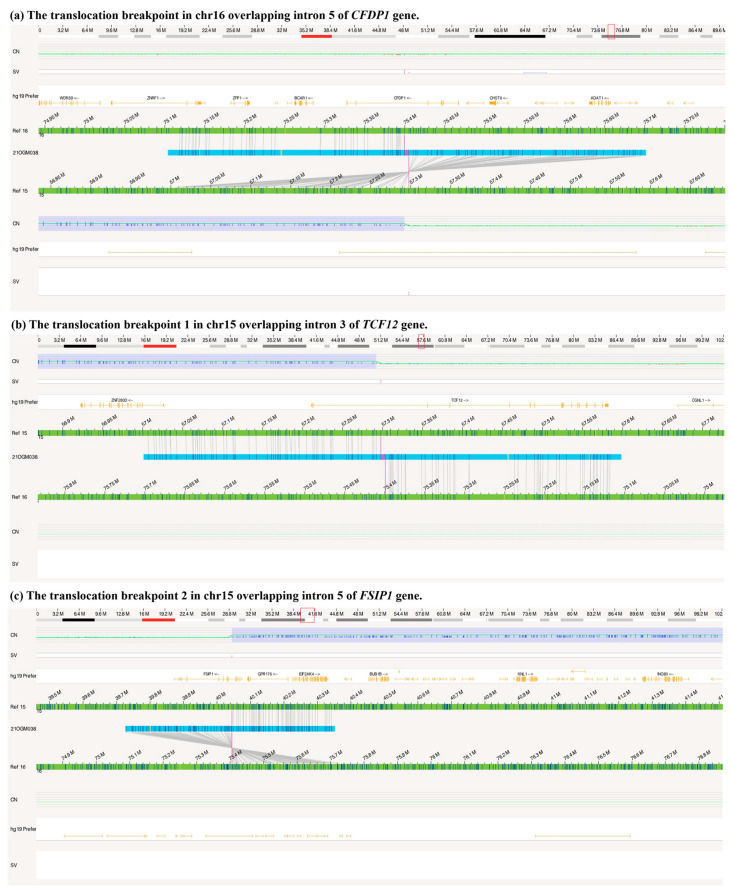
The genome view map showing the breakpoints in chromosomes 15 and 16. (**a**) the translocation breakpoints in chr16 overlapping intron 5 of *CFDP1* gene, with the blue block showing the copy number gain in 15q14q21.3. (**b**) the translocation breakpoint 1 in chr15 overlapping intron 3 of *TCF12* gene. (**c**) the translocation breakpoint 2 in chr15 overlapping intron 5 of *FSIP1* gene.

## Data Availability

Data supporting our findings is available upon reasonable request by the corresponding author.

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
