# Peer review of "The Application of Optical Genome Mapping (OGM) in Severe Short Stature Caused by Duplication of 15q14q21.3"

_genes, 2023, doi:10.3390/genes14051016_

Round 1

Reviewer 1 Report

COMMENTS TO AUTHORS

Introductory paragraph: The authors carried out genomic analyses on an individual with severe short stature (-3.41SDS) with abnormal skeletal dysplasia, mild to moderate tricuspid regurgitation, abnormal gait, and growth hormone deficiency 147 (GHD) using complementary analytical techniques. The techniques employed included optical genome mapping (OGM), whole exon sequencing (WES), copy number variation sequencing (CNV-seq), and karyotyping. Three techniques revealed two structural variants (SVs) in a 10.7-year-old boy, including a 17.27 Mb duplication of chromosome 15 (15q14-q21.3) ascertained by WES and CNV-Seq, and an insertion in chromosome 16 ascertained by karyotyping. However, OGM concluded that it was the duplication of 15q15.1q21.3 that was inversely inserted into 16q23.1 leading to two gene fusion events. Thus, there was an inter-translocation (inversion) between chromosomes 15 and 16, and a duplicative copy number of chromosome 15 of about 40.044 Mb-57.297Mb in size. The authors thus implicate 46, XY, der (16) ins (16;15) (q22; q21.3q15.1) dn as the possible cause of the phenotypes observed in the proband.

General comments:

Major points:

None

Minor points:

None

Specific comments:

Page 1 lines 22-23: the statement “In total, there were 14 patients with duplication of 22 15q15.1q21.3, and 42.9% of them were de-novo” creates the impression that the authors analyzed 14 patients. The author may want to rephrase this statement to indicate that 13 of such patients were from previous publications other than the research they are reporting here.

Page 1 line 36: replace “association” with “associated”.

Page 6 line 198: delete “with”.

Concluding paragraph: OGM mapping was great at confirming and refining the observations made by WES, CNV-Seq, and karyotyping. The complementary techniques employed by the authors, together with 13 already reported cases, suggest that the 46, XY, der (16) ins (16;15) (q22; q21.3q15.1) dn may be responsible for the phenotype observed in the proband.

Author Response

Specific comments:

Point 1: Page 1 lines 22-23: the statement “In total, there were 14 patients with duplication of 22 15q15.1q21.3, and 42.9% of them were de-novo” creates the impression that the authors analyzed 14 patients. The author may want to rephrase this statement to indicate that 13 of such patients were from previous publications other than the research they are reporting here.

Response 1: Thank you for your comments. We have rephrased the sentence into “A total of fourteen patients carried the duplication of 15q14q21.3 with thirteen previously reported and one from our center, and 42.9% of them were de-novo”, please see line 22-24, page 1.

Point 2: Page 1 line 36: replace “association” with “associated”.

Response 2: Thank you for your advice. We have changed “association” into “associated”, please see line 36, page 1.

Point 3: Page 6 line 198: delete “with”.

Response 3: Thank you for your advice. We have deleted the word “with”, please see line 208, page 6.

Reviewer 2 Report

The authors describe a case of a boy with severe short stature and mild skeletal dysplasia whose likely diagnosis was obtained by means of five combined molecular genetics techniques: exome sequencing, Sanger sequencing, copy number variation sequencing, karyotype and optical genome mapping. Through the combination of these techniques, it was possible to identify a duplication of 15q15.1q21.3 that was inversely inserted into 16q23.1, resulting in two fusion genes (TCF12-CFDP1 and FSIP1-CFDP1). The authors also searched previous literature and publicly available databases for cases of duplications within the 15q15.1q21.3 region, reviewing associated clinical phenotypes.

The manuscript has a value for the clinical community as the patient’s duplication and insertion had never been described before and displays the importance of application of optical genome mapping in the clinical setting for better characterization of similar rearrangements. 

Following are some points that need to be clarified. 

Major revisions needed:

-       The authors focus throughout the article on the importance of OGM for short stature only, while it is widely demonstrated that structural abnormalities cause a wide array of symptoms with short stature being a possible outcome among many. I would recommend that the authors change this overfocusing on short stature and comment on the application of the technique in all clinical settings in which a structural imbalance may be the underlying genetic cause. I therefore recommend reviewing lines 35-39, 49-50, 192-194, 251-252, 303-304 in the manuscript.

-       The precise genomic coordinates of the duplicated region are lacking, so it is difficult to identify the genes involved in the duplication. This piece of information should be added, together with the list of the genes involved in the duplication and what their roles are. The same goes for table 1, in which the localization of duplications is not precise, genomic coordinates should be included in the table.

-       It is not clear why the authors seem to be sure that the duplication-insertion found in the patient is the sole responsible of the clinical picture. This is likely but not certain due to the unique presentation and lack of functional demonstration of the impact of duplication and insertion. It should be clearly stated in the article that this is a speculation based on the available information. 

-       In lines 283-293 the authors speculate that FBN1, involved in the duplication, could explain the patient phenotype. However, they do not explain why the duplication of the gene should cause the same phenotype as missense or loss-of-function variants. The comparison between the clinical picture should be supported by known mechanisms of pathogenicity. Authors should describe in the results section what other genes are involved in the duplication and what their biological roles are, they should not focus only on FBN1.

Minor revisions:

-       In Table 1 the size of duplications should be described consistently in Mb or Kb and not in bp for some patients only. e.g. line 202, 65569 bp should be 65.6 Kb (it does not make sense not to approximate these values when the CNV described for the patient is written as 17.2 Mb). On the other hand, mentioned in the major revision section, precise genomic coordinates should be provided within this table whenever possible, so that a doublecheck on databases can be performed based on the manuscript’s data.

-       In the second paragraph of the introduction, it is not clear what the difference is between array comparative genomic hybridization and copy number variation (CNV) microarray. They are listed in the same sentence as if they were two different techniques: do they mean SNP-array when they mention CNV microarray? 

-       The term “additionally” is misused in the abstract section when describing the symptoms associated to previous cases of 15q15.1q21.3, since the only symptoms they refer to in this section are the neurological ones. 

Author Response

Point 1: The authors focus throughout the article on the importance of OGM for short stature only, while it is widely demonstrated that structural abnormalities cause a wide array of symptoms with short stature being a possible outcome among many. I would recommend that the authors change this overfocusing on short stature and comment on the application of the technique in all clinical settings in which a structural imbalance may be the underlying genetic cause. I therefore recommend reviewing lines 35-39, 49-50, 192-194, 251-252, 303-304 in the manuscript.

Response 1: Thank you for your comments and we agree with you. The potential application of OGM in exploring the genetic etiology of patients with clinical syndrome were stated in revised manuscript, please see lines 35-39, 48-50, 202-204, 219, 259-261, 340-341 in the revised manuscript.

Point 2: The precise genomic coordinates of the duplicated region are lacking, so it is difficult to identify the genes involved in the duplication. This piece of information should be added, together with the list of the genes involved in the duplication and what their roles are. The same goes for table 1, in which the localization of duplications is not precise, genomic coordinates should be included in the table.

Response 2: Thank you for your comments. The precise genomic coordinates of the structural variations in our proband have been added in the revised manuscript, please see lines 196-201, page 6. Also, the genomic coordinates of structural variations previously reported were added in Table 1.  In addition, there were 309 genes involved in duplication of 15q14q21.3 and the list of the genes was presented as Supplementary material, which shows the corresponding OMIM code.

Further, GO enrichment and KEGG pathway analysis of the genes in the duplication were realized through Metascape (http://metascape.org). GO analyses revealed that these genes were mainly enriched in the following processes: microtubule cytoskeleton organization, phosphatidylglycerol acyl-chain remodeling, chromosome segregation, vesicle organization, etc (Supplementary material, Figure 1). The KEGG pathways enrichment showed that these genes were mainly enriched in three pathways: ovarian steroidogenesis, thyroid hormone synthesis, and arginine and proline metabolism (Supplementary material, Figure 2). We further added the content in the Methods section (please see lines 108-110) and Results section (please see lines 174-181).   

Supplementary material (see figures in the attachment)

Figure 1. Top 20 clusters of GO enrichment for the genes in duplication of 15q14q21.3.

Figure 2. Top 3 clusters of KEGG pathway enrichment for the genes in duplication of 15q14q21.3.

Point 3: It is not clear why the authors seem to be sure that the duplication-insertion found in the patient is the sole responsible of the clinical picture. This is likely but not certain due to the unique presentation and lack of functional demonstration of the impact of duplication and insertion. It should be clearly stated in the article that this is a speculation based on the available information.

Response 3: Thank you for your comments. First, according to a joint consensus recommendation of the American College of Medical Genetics and Genomics (ACMG) and the Clinical Genome Resource (ClinGen), the chromosomal structural variation involved 17.2 Mb in our proband, much larger than 5Mb, was considered to be a pathogenic variation [1]. Second, we reviewed the literature and searched databases to find out the function and phenotypes of all genes involved in duplication of 15q14q21.3, and concluded that the phenotypes of the patient may be relatively consistent with that caused by FBN1. Moreover, duplication of FBN1 gene was a possible cause of skeletal abnormalities in the patient with 15q15.3q21.2 duplication has been reported by Yuan et al [2], but the specific mechanism remained unknown. In conclusion, we hypothesized that the duplication of 15q14-21.3 was the pathogenic cause for the patient’s phenotypes, and FBN1 was a candidate pathogenic gene. However, due to the lack of functional verification, we cannot be sure that the duplication of FBN1 was responsible for phenotypes of our proband.   Thus, we have restated in the revised manuscript that this is a speculation based on the available information, please see lines 202-204, 287-289 and 316-318.

Reference

[1] Riggs, E.R., et al., Technical standards for the interpretation and reporting of constitutional copy-number variants: a joint consensus recommendation of the American College of Medical Genetics and Genomics (ACMG) and the Clinical Genome Resource (ClinGen). Genet Med, 2020. 22(2): p. 245-257.

[2] Yuan, H., et al., A rare de novo interstitial duplication of 15q15.3q21.2 in a boy with severe short stature, hypogonadism, global developmental delay and intellectual disability. Mol Cytogenet, 2016. 9: p. 2.

Point 4: In lines 283-293 the authors speculate that FBN1, involved in the duplication, could explain the patient phenotype. However, they do not explain why the duplication of the gene should cause the same phenotype as missense or loss-of-function variants. The comparison between the clinical picture should be supported by known mechanisms of pathogenicity. Authors should describe in the results section what other genes are involved in the duplication and what their biological roles are, they should not focus only on FBN1.

Response 4: Thank you for your comments. Disease-causing mutations of FBN1 disrupting heparin binding by TB5 can resulted in Weill-Marchesani syndrome (WMS) or Acromicric (AD) and Geleophysic Dysplasias (GD) [1]. Moreover, duplication of FBN1 gene was a possible cause of skeletal abnormalities in the patient with 15q15.3q21.2 duplication has been reported by Yuan et al [2], but the specific mechanism remained unknown. Indeed, the phenotypes of FBN1, located in 15q21.1, are related to severe short stature, skeletal dysplasia, toe walking, normal intelligence, and distinctive facial features, consistent with those of our proband. In addition, we had searched the function and phenotypes of all genes in 15q14q21.3 and only FBN1 gene could explain the complex clinical phenotypes of our proband. Therefore, we speculate that the structural variation may be responsible for the clinical phenotypes of our proband and FBN1 was the candidate gene. We have added the content in the Discussion section, please see lines 303-318.

Also, all genes involved in duplication of 15q14q21.3 and their roles were described in Result section (lines 174-181) and the information of them was offered as supplementary materials. In addition, partial genes were discussed in Discussion section, please see lines 322-330.

Reference:

[1] Murray, P.G., P.E. Clayton, and S.D. Chernausek, A genetic approach to evaluation of short stature of undetermined cause. Lancet Diabetes Endocrinol, 2018. 6(7): p. 564-574.

[2] Yuan, H., et al., A rare de novo interstitial duplication of 15q15.3q21.2 in a boy with severe short stature, hypogonadism, global developmental delay and intellectual disability. Mol Cytogenet, 2016. 9: p. 2.

Minor revisions:

Point 5: In Table 1 the size of duplications should be described consistently in Mb or Kb and not in bp for some patients only. e.g. line 202, 65569 bp should be 65.6 Kb (it does not make sense not to approximate these values when the CNV described for the patient is written as 17.2 Mb). On the other hand, mentioned in the major revision section, precise genomic coordinates should be provided within this table whenever possible, so that a doublecheck on databases can be performed based on the manuscript’s data.

Response 5: Thank you for your comments. We have revised “65569 bp” into “65.6Kb”, please see line 212 and Table 1. Also, the genomic coordinates of structural variations previously reported were added in Table 1.

Point 6:  In the second paragraph of the introduction, it is not clear what the difference is between array comparative genomic hybridization and copy number variation (CNV) microarray. They are listed in the same sentence as if they were two different techniques: do they mean SNP-array when they mention CNV microarray?

Response 6: Thank you for your comments. Array comparative genomic hybridization (aCGH) is one of copy number variation (CNV) microarrays. We have deleted “Array comparative genomic hybridization (aCGH)” in the revised manuscript, please see lines 40-41.

Point 7: The term “additionally” is misused in the abstract section when describing the symptoms associated to previous cases of 15q15.1q21.3, since the only symptoms they refer to in this section are the neurological ones.

Response 7: Thank you for your advice. We have changed the word “Additionally” into “in addition”, please see line 23, page 1.
